# Evolving Pareto-Optimal Actor-Critic Algorithms for Generalizability and Stability

## Abstract

Generalizability and stability are two key objectives for operating reinforcement learning (RL) agents in the real world. Designing RL algorithms that optimize these objectives can be a costly and painstaking process. This paper presents MetaPG, an evolutionary method for automated design of actor-critic loss functions. MetaPG explicitly optimizes for generalizability and performance, and implicitly optimizes the stability of both metrics. We initialize our loss function population with Soft Actor-Critic (SAC) and perform multi-objective optimization using fitness metrics encoding single-task performance, zero-shot generalizability to unseen environment configurations, and stability across independent runs with different random seeds. On a set of continuous control tasks from the Real-World RL Benchmark Suite, we find that our method, using a single environment during evolution, evolves algorithms that improve upon SAC's performance and generalizability by 4% and 20%, respectively, and reduce instability up to 67%. Then, we scale up to more complex environments from the Brax physics simulator and replicate generalizability tests encountered in practical settings, such as different friction coefficients. MetaPG evolves algorithms that can obtain 10% better generalizability without loss of performance within the same meta-training environment and obtain similar results to SAC when doing cross-domain evaluations in other Brax environments. The evolution results are interpretable; by analyzing the structure of the best algorithms we identify elements that help optimizing certain objectives, such as regularization terms for the critic loss.

## 1 Introduction

Two key bottlenecks for the deployment of reinforcement learning (RL) in the real world are failing to generalize beyond the training distribution and unstable training. Both are common in practical settings and constitute two important aspects of RL robustness. On one hand, many real-world environments present themselves in multiple configurations (e.g., different sizes, structure, context, properties) and practitioners expect zero-shot generalization when facing new configurations in robot manipulation (Ibarz et al., 2021), navigation (Chiang et al., 2019), energy systems (Perera & Kamalaruban, 2021), and fluid dynamics (Garnier et al., 2021). On the other hand, real-world elements such as stochastic dynamics should not result in unstable learning behaviors that lead to undesired performance drops. Even for state-of-the-art RL algorithms, zero-shot generalization and instability are considerable challenges (Henderson et al., 2018; Dulac-Arnold et al., 2021).

Improving generalizability has been addressed by learning or encoding inductive biases in RL algorithms (Raileanu & Fergus, 2021; Vlastelica et al., 2021). Gains mostly come by manually modifying existing algorithms (Cobbe et al., 2019; Igl et al., 2019; Cobbe et al., 2021). As RL design for real-world environments tends to be empirical (Andrychowicz et al., 2020; Linke et al., 2020), finding the suitable algorithmic changes that benefit generalizability might take many iterations, especially if performance loss must be avoided (see Figure 1). As environments become more complex and inductive biases become environment-specific, the cost of human-driven design might be too expensive when optimizing for generalizability (Zhao et al., 2019), let alone when optimizing for both performance and generalizability (Hessel et al., 2019), which are just two objectives we consider but there are many more. In addition, stable algorithms that show consistent

performance and generalizability across independent runs leads to higher algorithm reusability within the same environment and across environments. We argue that optimizing generalizability and stability in addition to performance in RL builds the case for automating algorithm design and speeding up the process of RL algorithm discovery. Automated Machine Learning or AutoML (Hutter et al., 2019) has proven to be a successful tool for supervised learning problems (Vinyals et al., 2016; Zoph et al., 2018; Real et al., 2019; Finn et al., 2017), and it has been recently applied in the context of RL for automating loss function search (Oh et al., 2020; Xu et al., 2020; Co-Reyes et al., 2021; Bechtle et al., 2021; He et al., 2022; Lu et al., 2022).

This paper proposes MetaPG (see Figure 2), a method that evolves a population of actor-critic RL algorithms (Sutton & Barto, 2018), identified by their loss functions, with the goal of increasing single-task performance, zero-shot generalizability, and stability across independent runs. Loss functions are represented symbolically as directed acyclic computation graphs and two independent fitness scores are used to encode performance and generalizability, respectively. A measure for stability is accounted for in both objectives and the multi-objective ranking algorithm NSGA-II (Deb et al., 2002) is used to identify the best algorithms. Compared to manual design, this strategy allows us to explore the algorithm space more efficiently by automating search operations. MetaPG finds algorithm improvement directions that jointly optimize both objectives until it obtains a Pareto Front of loss functions that maximizes fitness with respect to each objective, approximating the underlying tradeoff between them. On one end of the Pareto Front we obtain algorithms that perform well in the training task (beneficial when faster learning is required and overfitting is not a concern), on the other end we find algorithms that generalize better to unseen configurations, and in between there are algorithms that interpolate between both behaviors.

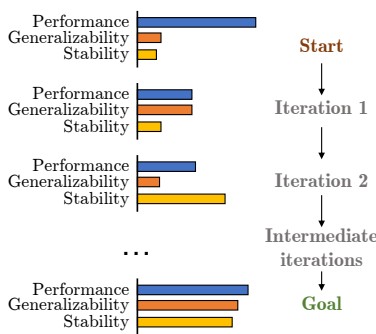

Figure 1: In many practical contexts, designing the RL agent that achieves the right performance, generalizability, and stability is an empirical and costly iteration-based process.

To evaluate MetaPG, we run experiments using, first the zero-shot generalizability benchmark from the Real-World RL environment suite (Dulac-Arnold et al., 2021); and second, the Ant and Humanoid environments from Brax physics simulator Freeman et al. (2021), where we simulate perturbations like mass changes, different friction coefficients, and joint torques. We warm-start the evolution with a graph-based representation of Soft Actor-Critic (SAC) (Haarnoja et al., 2018), and demonstrate that our method is able to evolve a Pareto Front of multiple actor-critic algorithms that outperform SAC and increase its performance and generalizability up to 4% and 20%, respectively, and decrease instability up to 67%. In the Brax environments, the evolved algorithms outperform SAC by 15% and 10% in performance and generalizability, respectively. Instability is reduced 23% in that case. Furthermore, we observe that algorithms evolved in Ant show minimal generalizability loss when transferred to Humanoid and vice versa. Finally, by inspecting the graphs of the evolved algorithms, we interpret which substructure drive the gains. For instance, we find that MetaPG evolves loss functions that remove the entropy term in SAC to increase performance.

In summary, this paper makes three main contributions:

1. A method that combines multi-objective evolution with a search language representing actor-critic RL algorithms as graphs, which can discover new loss functions over a set of different objectives.

2. A formulation of stability-adjusted scores that explicitly optimize for performance and generalizability and implicitly encourage stability.

3. A dataset[1] of Pareto-optimal actor-critic loss functions which outperform baselines like SAC on multiple objectives. This dataset may be further analyzed to understand how algorithmic changes affect the tradeoff between different objectives.

---

[1]The dataset can be found at: https://github.com/authors2022/dataset

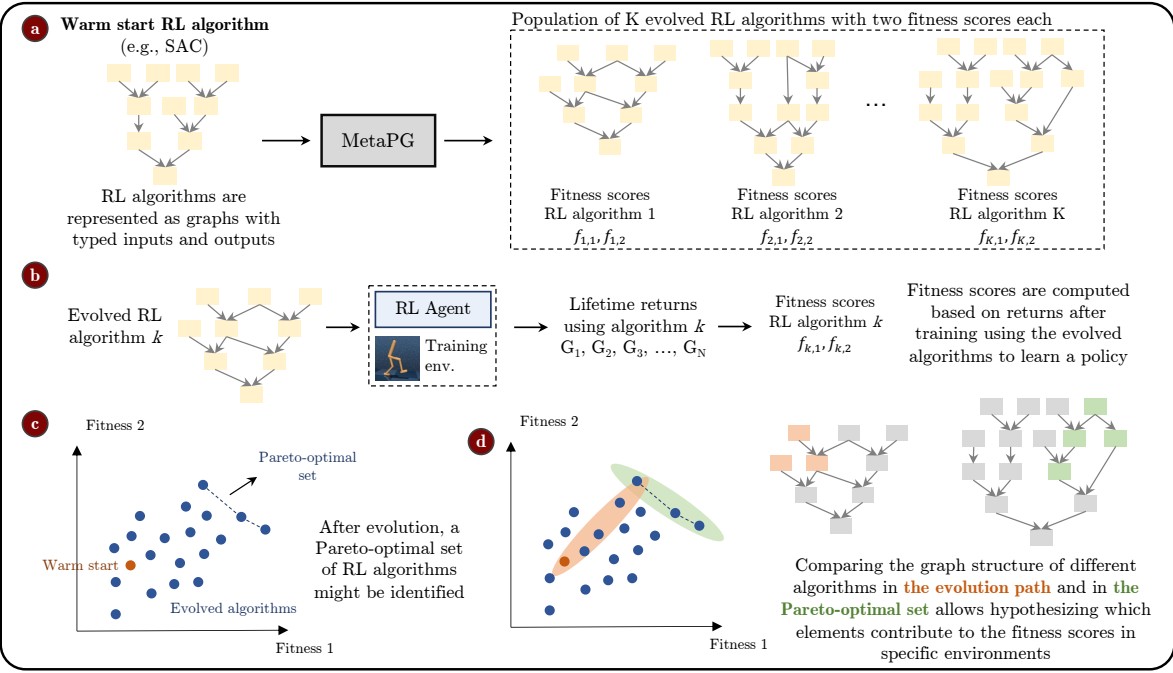

Figure 2: MetaPG overview, example with two fitness scores encoding two RL objectives. **(a)** The method starts by taking a warm-start RL algorithm with its loss function represented in the form of a directed acyclic graph. MetaPG consists of a meta-evolution process that, after initializing algorithms to the warm-start, discovers a population of new algorithms. **(b)** Each evolved graph is evaluated by training an agent following the algorithm encoded by it, and then computing two fitness scores based on the training outcome. **(c)** After evolution, all RL algorithms can be represented in the fitness space and a Pareto-optimal set of algorithms can be identified. **(d)** Identifying which graph substructures change across the algorithms in the Pareto set reveals which operations are useful for specific RL objectives. MetaPG can be scaled to more than two RL objectives.

## 2 Related Work

**Generalizability in RL**    One of the aspects of RL robustness is generalizability (Kirk et al., 2022; Xu et al., 2022). Increasing generalizability has been addressed by means of environment randomization (Tobin et al., 2017; Peng et al., 2018; Akkaya et al., 2019). Other authors have shown that removing or adding certain algorithmic components impacts generalizability (e.g. using batch normalization (Cobbe et al., 2019), adding elements to rewards (Chen, 2020), or using regularizers (Igl et al., 2019; Cobbe et al., 2019)). Others directly achieve generalizability gains by modifying existing actor-critic RL algorithms (Raileanu & Fergus, 2021; Cobbe et al., 2021). Vlastelica et al. (2021) propose a hybrid architecture combining a neural network with a shortest path solver to find better inductive biases that improve generalizing to unseen environment configurations. We automate the search of algorithmic changes for actor-critic algorithms using generalizability as one of the search metrics.

**Stability in RL**    Achieving stable behaviors during training is essential in many domains, particularly in control applications (Ibarz et al., 2021; Azar et al., 2021). It has been shown that randomness can play a substantial role in the outcome of a training run (Henderson et al., 2018). Stable learning has been sought by means of algorithmic innovation (Haarnoja et al., 2018; Fox et al., 2015; Bao et al., 2021; Jiang et al., 2021). New stable algorithms have been mainly developed after looking into stability in isolation; in this work we focus on stability as one of the three objectives to simultaneously optimize.

**Optimizing RL components**    Automated RL or AutoRL seeks to meta-learn RL components (Parker-Holder et al., 2022), such as RL algorithms (Co-Reyes et al., 2021; Kirsch et al., 2020; Oh et al., 2020; Bechtle et al., 2021), their hyperparameters (Zhang et al., 2021; Hertel et al., 2020; Xu et al., 2018), policy/neural

network (Gaier & Ha, 2019; Miao et al., 2021), or the environment (Ferreira et al., 2021; Gur et al., 2021; Dennis et al., 2020; Florensa et al., 2018; Volz et al., 2018; Faust et al., 2019). This work evolves RL loss functions and leaves other elements of the RL problem out of the scope. We focus on the RL loss function given its interaction with all elements in a RL problem: states, actions, rewards, and the policy.

**Evolutionary AutoML**  Neuro-evolution introduced evolutionary methods in the context of AutoML (Miller et al., 1989; Stanley & Miikkulainen, 2002), including neural network architecture search (Stanley et al., 2009; Jozefowicz et al., 2015; Real et al., 2019). In the RL context evolution searched for policy gradients (Houthooft et al., 2018) and value iteration losses (Co-Reyes et al., 2021). Our work is also related to the field of genetic programming, in which the goal is to discover computer code (Koza, 1994; Real et al., 2020; Co-Reyes et al., 2021). In this work we use a multi-objective evolutionary method to discover new RL algorithms, specifically actor-critic algorithms (Sutton & Barto, 2018), represented as graphs.

**Learning RL loss functions**  Loss functions play a central role in RL algorithms and are traditionally designed by human experts. Recently, several lines of work propose to view RL loss functions as tunable objects that can be optimized automatically (Parker-Holder et al., 2022). One popular approach is to use neural loss functions whose parameters are optimized via meta-gradient (Kirsch et al., 2020; Bechtle et al., 2021; Oh et al., 2020; Xu et al., 2020; Lu et al., 2022). An alternative is to use symbolic representations of loss functions and formulate the problem as optimizing over a combinatorial space. One example is (Alet et al., 2019), which represents extrinsic rewards as a graph and optimizes it by cleverly pruning a search space. Learning value-based RL loss functions by means of evolution was first proposed by Co-Reyes et al. (2021), and was applied to solving discrete action problems. He et al. (2022) propose a method to evolve auxiliary loss functions which complemented predefined standard RL loss functions. MetaPG focuses on continuous control problems and searches for complete symbolic loss functions of actor-critic algorithms.

## 3 Methods

We represent actor-critic loss functions (policy loss and critic loss) as directed acyclic graphs and use an evolutionary algorithm to evolve a population of graphs, which are ranked based on their fitness scores. The population is warm-started with known algorithms such as SAC and undergoes mutations over time. Each graph's fitnesses are measured by training from scratch an RL agent with the corresponding loss function and encode performance and generalizability as explicit objectives, and stability as a third objective defined implicitly. We use the multi-objective evolutionary algorithm NSGA-II (Deb et al., 2002) to jointly optimize all fitness scores until growing a Pareto-optimal set of graphs or Pareto Front. Algorithm 1 summarizes the process; `Offspring` and `RankAndSelect` are NSGA-II subroutines.

Section 3.1 provides RL algorithm graph representation details. The main logic of MetaPG is contained in the evaluation routine, which computes fitness scores (Section 3.2) and employs several techniques to speed up the evolution and evaluation processes (Section 3.3). See Appendix B for further implementation details.

### 3.1 RL algorithm representation

MetaPG encodes loss functions as graphs consisting of typed nodes sufficient to represent a wide class of actor-critic algorithms. Compared to the prior value-based RL evolutionary search method introduced by Co-Reyes et al. (2021), MetaPG's search space greatly expands on it and adds input and output types to manage the search complexity. As a representative example, Figure 8 in Appendix D presents the encoding for SAC that we use in this paper. In our experiments we limit the number of nodes per graph to 60 and 80, which can represent approximately $10^{300}$ and $10^{400}$ graphs, respectively (see Appendix A.2). Nodes in the graph encode loss function inputs, operations, and loss function outputs. The inputs include elements from transition tuples, constants such as the discount factor $\gamma$, a policy network $\pi$, and multiple critic networks $Q_i$. Operation nodes support intermediate algorithm instructions such as basic arithmetic neural network operations. Then, outputs of the graphs correspond to the policy and critic losses. The gradient descent minimization process takes these outputs and computes their gradient with respect to the respective network parameters. In Appendix A.1 we provide a full description of the search language and nodes considered.

---

**Algorithm 1** MetaPG Overview

---

**Input:** Training environments $\mathcal{E}$
**Initialize**: Initialize population $P_0$ of loss function graphs (random initialization or bootstrap with an algorithm such as SAC).

1: **for** $L$ in $P_0$ **do** $L.score \leftarrow \text{Eval}(L, \mathcal{E})$
2: **end for**
3: $Q_0 \leftarrow \text{Offspring}(P_0)$                  ▷ NSGA-II
4: **for** $L$ in $Q_0$ **do** $L.score \leftarrow \text{Eval}(L, \mathcal{E})$
5: **end for**
6: **for** $t = 1$ **to** $G$ **do**
7:     $R \leftarrow P_{t-1} \bigcup Q_{t-1}$
8:     $P_t \leftarrow \text{RankAndSelect}(R)$           ▷ NSGA-II
9:     $Q_t \leftarrow \text{Offspring}(P_t)$               ▷ NSGA-II
10:     **for** $L$ in $Q_t$ **do** $L.score \leftarrow \text{Eval}(L, \mathcal{E})$
11:     **end for**
12: **end for**
13: **Output:** Pareto Front of all loss function graphs.

---

MetaPG's search language supports both on-policy and off-policy algorithms; however, in this paper we focus on off-policy algorithms given their better sample efficiency.

## 3.2 Fitness scores

This work focuses on optimizing single-task performance, zero-shot generalizability, and stability across independent runs with different random seeds. The process to compute the fitness scores is depicted in Figure 3. We use $N$ random seeds and a set of environments $\mathcal{E}$, which comprises multiple instances of the same environment class, including a training instance $E_{train} \in \mathcal{E}$. For example, $\mathcal{E}$ is the set of all RWRL Cartpole environments with different pole lengths (0.1 meters to 3 meters in 10-centimeter intervals), and $E_{train}$ corresponds to an instance with a specific pole length (1 meter). The first step (see Figure 3a) is to compute raw performance and generalizability scores for each individual seed. Using $E_{train}$ and seed $k$ to train a policy $\pi_k$, the performance score $f_{perf_k}$ is the average evaluation return on the training environment configuration:

$$f_{perf_k} = \frac{1}{N_{eval}} \sum_{n=1}^{N_{eval}} G_n(\pi_k, E_{train}), \tag{1}$$

where $G_n$ corresponds to the normalized return for episode $n$ given a policy and an environment instance, and $N_{eval}$ is the number of evaluation episodes. Algorithms that learn faster in the training environment and overfit to it obtain higher performance scores. The generalizability score $f_{gen_k}$ is in turn computed as the average evaluation return of the policy trained on $E_{train}$ over the whole range of environment configurations. We emphasize that the policy is trained on a single environment configuration (for example 1.0 meter pole length) and then is evaluated in a zero-shot fashion to new unseen environment configurations:[2]

$$f_{gen_k} = \frac{1}{|\mathcal{E}| N_{eval}} \sum_{E \in \mathcal{E}} \sum_{n=1}^{N_{eval}} G_n(\pi_k, E) \tag{2}$$

We look for stable training results for both performance and generalizability. To that end, once we have independent raw scores for each seed ($N$ different performance and generalizability scores) we define the *stability-adjusted* scores (see Figure 3b) as

$$\tilde{f} = \mu(\{f_n\}_{n=1}^N) - \kappa \cdot \sigma(\{f_n\}_{n=1}^N) \tag{3}$$

---

[2]More precisely, it should be $E \in \mathcal{E} \setminus \{E_{train}\}$. In practice, we find this makes no significant difference in the metric because the number of test configurations is normally around 30.

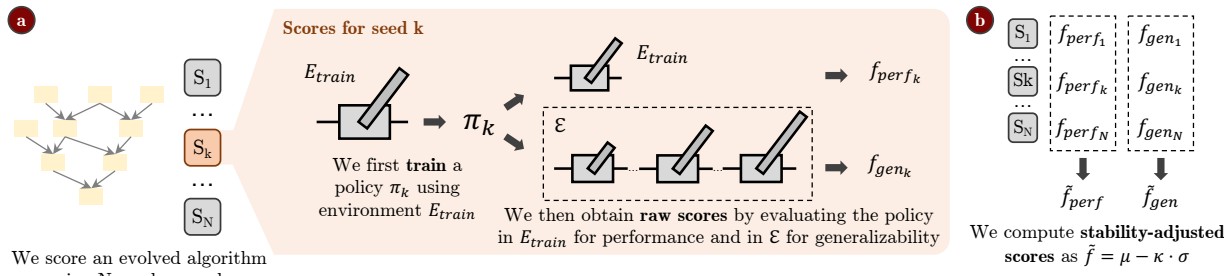

Figure 3: Process to compute fitness scores. **(a)** Independently for each seed from a set of $N$ seeds, we first train a policy $\pi_k$ using environment $E_{train}$ and seed $k$. During training the policy is allowed to take stochastic actions. Then, we evaluate $\pi_k$ deterministically on that same environment $E_{train}$ to get a raw performance score $f_{perf_k}$, and on a set of environments $\mathcal{E}$ (same environment with different configurations; e.g., different pole lengths as shown) to get a raw generalizability score $f_{gen_k}$. **(b)** We compute stability-adjusted fitness scores $\tilde{f}_{perf}$ and $\tilde{f}_{gen}$ by aggregating raw scores from each seed using Equation 3.

where $f$ is a score (performance or generalizability), $f_n$ denotes the score for seed $n$; $\mu$ and $\sigma$ are the mean and standard deviation across the $N$ seeds, respectively; and $\kappa$ is a penalization coefficient. The final fitness of a graph is the tuple $(\tilde{f}_{perf}, \tilde{f}_{gen})$.

## 3.3 Evolution details

**Mutation**  The population is initialized with a warm-start RL algorithm; all individuals are copies of this algorithm's graph at the beginning. Once the population is initialized, individuals undergo mutations that change the structure of their respective graphs. Specifically, mutations consist of either replacing one or more nodes in the graph or changing the connectivity for one edge. The specific number of nodes that are affected by mutation is randomly sampled for each different individual; see Appendix B for more details.

**Operation consistency**  To prevent introducing corrupted child graphs into the population, MetaPG checks operation consistency, i.e., for each operation, it makes sure the shapes of the input tensors are valid and compatible, and computes the shape of the output tensor. These shapes and checks are propagated along the computation graph.

**Hashing**  To avoid repeated evaluations, MetaPG hashes (Real et al., 2020) all graphs in the population. Once the method produces a child graph and proves its consistency, it computes a hash value and, in case of a cache hit, reads the fitness scores from the cache. Since only the gradient of a loss function matters during training, we hash a graph by computing the corresponding loss function's gradient on synthetic inputs.

**Hurdle evaluations**  We carry out evaluations for different individuals in the population in parallel, while evaluating across seeds for one algorithm is done sequentially. To prevent spending too many resources on algorithms that are likely to yield bad policies, MetaPG uses a simple hurdle environment (Co-Reyes et al., 2021) and a number of hurdle seeds. We first evaluate the algorithm on the hurdle environment for each hurdle seed, and only proceed with more complex and computationally expensive environments if the resulting policy performs above a certain threshold on the hurdle environment.

## 4 Results

This section aims to answer the following questions:

1. Is MetaPG capable of evolving algorithms that improve upon performance, generalizability, and stability in different practical settings?

2. How well do discovered algorithms do in environments different from those used to evolve them?

3. Are the evolutionary results interpretable?

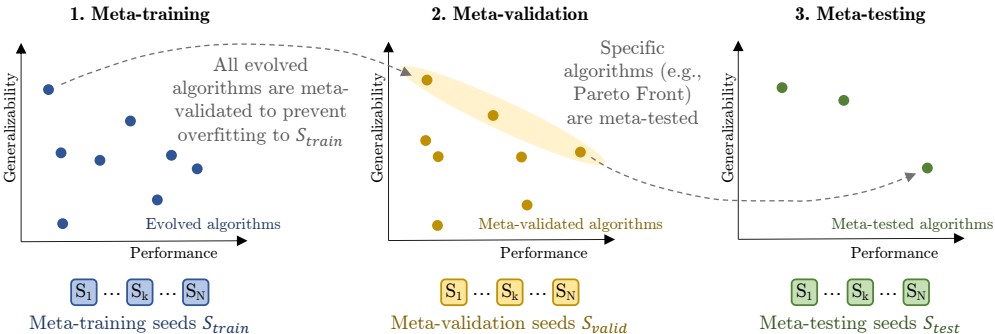

Figure 4: Running an experiment with MetaPG is divided into three phases. **1. Meta-training**: we evolve a population of algorithms using a set of random seeds $S_{train}$ to compute scores. **2. Meta-validation**: to prevent overfitting, we reevaluate the scores of all algorithms in the population using a different set of random seeds $S_{valid}$. **3. Meta-testing**: specific algorithms such as those in the Pareto Front are tested in different environments using a third set of random seeds $S_{test}$.

We run the experiments following the process represented in Figure 4; we divide them into meta-training, meta-validation, and meta-testing phases. Each phase relies on a different set of $N$ random seeds: $S_{train}$, $S_{valid}$, and $S_{test}$, respectively. Each MetaPG run begins with the meta-training phase, which consists of the evolution process described in the previous section. The result of this phase is a population of evolved algorithms, each with a pair of meta-training fitness scores. Since the evolution process is non-deterministic, we run each experiment multiple times without configuration changes and aggregate all resulting populations into one single larger population.

Then, to avoid selecting algorithms that overfit to the set of seeds $S_{train}$, we reevaluate all algorithms in the population with a different set of seeds $S_{valid}$; this corresponds to the meta-validation phase, which provides updated fitness scores for all algorithms. Finally, to assess the fitness of specific algorithms (e.g., the Pareto Front) when deployed in different environments, we use a third set of seeds $S_{test}$ that provides realistic fitness scores in the new environments; this corresponds to the meta-testing phase. In our analyses of the results, we focus on the set of meta-validated algorithms and then meta-test some of them.

## 4.1 Training setup

**Training environments**   We use as training environments: Cartpole and Walker from the RWRL Environment Suite (Dulac-Arnold et al., 2021), Gym Pendulum, and Ant and Humanoid from the Brax physics simulator (Freeman et al., 2021). We define different instances of these environments by varying the pole length in Cartpole, the thigh length in Walker, the pendulum length and mass in Pendulum, and, to mimic a practical setting, the mass, friction coefficient, and torque in Ant and Humanoid. See Appendix C for the specifics.

**Meta-training details**   The population and maximum graph size consist of 100 individuals and 80 nodes in the Brax environments, respectively, and 1,000 individuals and 60 nodes in the rest of the environments. All are initialized using SAC as a warm-start (see Appendix D). For RL algorithm evaluation, we use 10 different seeds $S_{train}$ and fix the number of evaluation episodes $N_{eval}$ to 20. In the case of Brax, since training takes longer, we use 4 different seeds but increase $N_{eval}$ to 32. We meta-train using 100 TPU 1x1 v2 chips for 4 days in the case of Brax environments (∼200K and ∼50K evaluated graphs in Ant and Humanoid, respectively), and using 1,000 CPUs for 10 days in the rest of environments (∼100K evaluated graphs per experiment). In all cases we normalize the fitness scores to the range [0, 1]. We set $\kappa = 1$ in equation 3. Additional details are in Appendix B.

**Meta-validation details**   During meta-validation, we use a set of 10 seeds $S_{valid}$, disjoint with respect to $S_{train}$. In the case of Brax environments, we use 4 meta-validation seeds. Same applies during meta-testing. In each case, we use a number of seeds that achieve a good balance between preventing overfitting and having

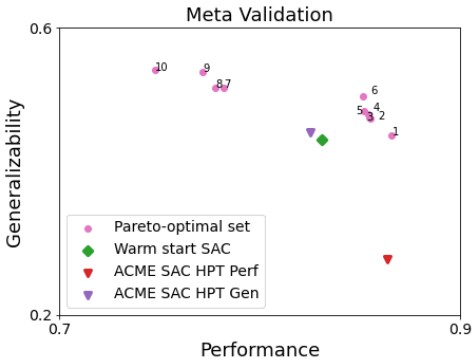
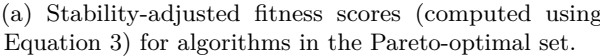

(a) Stability-adjusted fitness scores (computed using Equation 3) for algorithms in the Pareto-optimal set.

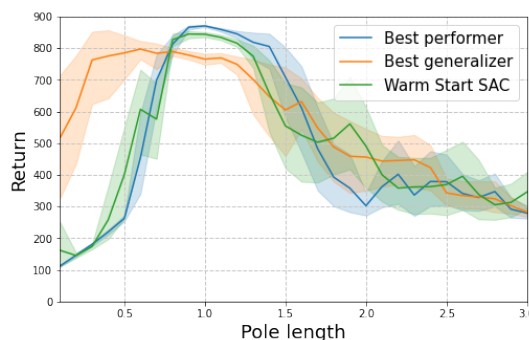

(b) Average return and standard deviation across seeds when evaluating trained policies in multiple RWRL Cartpole instances with different pole lengths (Training configuration is 1.0, see Appendix C).

Figure 5: Evolution results (meta-validation across 10 different seeds) alongside the warm-start algorithm (SAC), and the hyperparameter-tuned ACME SAC when using the RWRL Cartpole environment for training. We show the Pareto Front of algorithms that results after merging the 10 populations corresponding to the 10 repeats of the experiment. The best performer and best generalizer correspond to the algorithms with the highest stability-adjusted performance and generalizability scores, respectively, according to Equations 1, 2, and 3.

affordable evaluation time. The value of $N_{eval}$ during meta-validation and meta-testing matches the one used in meta-training.

**Hyperparameter tuning** We use the same fixed hyperparameters during all meta-training. Algorithms are also meta-validated using the same hyperparameters. In the case of Brax environments, we do hyperparameter-tune the algorithms during meta-validation; additional details can be found in Appendix G.5. We also hyperparameter-tune all baselines we compare our evolved algorithms against.

**RL Training details** The architecture of the policies corresponds to two-layer MLPs with 256 units each. Additional training details are presented in Appendix E.

## 4.2 Optimizing performance, generalizability, and stability in RWRL Environment Suite

We apply MetaPG to RWRL Cartpole and compare the evolved algorithms in the meta-validated Pareto Front with the warm-start SAC and ACME SAC (Hoffman et al., 2020) (Figure 5a). When running ACME SAC we first do hyperparameter tuning and pick the two configurations that lead to the best stability-adjusted performance and the best stability-adjusted generalizability (ACME SAC HPT Perf and ACME SAC HPT Gen, respectively). In Table 1, we show numeric values for each of the three metrics. In the case of stability, we show a measure of instability represented as the change in standard deviation compared to the warm-start (i.e., warm-start has the value 1.0). We independently measure instability with respect to performance and generalizability.

The results from Table 1 show that MetaPG discovers RL algorithms that improve upon the warm-start's and ACME SAC's performance, generalizability, and stability in the same environment we used during evolution. Compared to the warm-start, the best performer achieves a 4% improvement in the stability-adjusted performance score (from 0.836 to 0.868), the best generalizer achieves a 20% increase in the stability-adjusted generalizability score (from 0.460 to 0.551), and the selected algorithm in the Pareto-optimal set (Pareto point 6) achieves a 2% and a 12% increase in both stability-adjusted performance and stability-adjusted generalizability, respectively. Then, in terms of the stability objective, the best performer reduces performance instability by 67% and the best generalizer achieves a reduction of 30% for generalizability instability.

The gains in generalizability and stability are substantial when comparing the results to hyperparameter-tuned ACME SAC. The best generalizer achieves a 15% increase in stability-adjusted generalizability compared to ACME SAC tuned for such metric. The instability in the hyperparameter-tuned SAC is twice as high (1.48

Table 1: We compare three algorithms in the Pareto Front with SAC (warm-start and hyperparameter-tuned ACME SAC) using metrics obtained in the RWRL Cartpole environment: average performance and generalizability, stability-adjusted performance and generalizability scores, and measure of instability (standard deviation $\sigma$ divided by the warm-start's $\sigma_{WS}$). We compute these metrics across 10 seeds and the best result in a *column* is **bolded**. [†]In contrast to performance and generalizability, the lower the instability the better.

| | Performance | | Generalizability | | Instability[†] $(\sigma/\sigma_{WS})$ | |
| :--- | :---: | :---: | :---: | :---: | :---: | :---: |
| **RL Algorithm** | $f_{perf}$ | $\tilde{f}_{perf}$ | $f_{gen}$ | $\tilde{f}_{gen}$ | Perf. | Gen. |
| **Pareto point 1: Best performer** | **0.871** | **0.868** | 0.475 | 0.459 | 0.33 | **0.59** |
| **Pareto point 6** | 0.854 | 0.852 | 0.531 | 0.514 | 0.22 | 0.63 |
| **Pareto point 10: Best generalizer** | 0.770 | 0.756 | **0.570** | **0.551** | 1.55 | 0.70 |
| **Warm-start SAC** | 0.845 | 0.836 | 0.487 | 0.460 | *1.0* | *1.0* |
| **ACME SAC HPT Perf** | 0.865 | 0.864 | 0.372 | 0.312 | **0.11** | 2.22 |
| **ACME SAC HPT Gen** | 0.845 | 0.833 | 0.518 | 0.478 | 1.33 | 1.48 |

vs. 0.70, as shown in Table 1). In terms of performance, the best performer achieves a slightly better result compared to SAC hyperparameter-tuned for performance.

We report the complete results in Appendix F.1 and repeat the same experiments in RWRL Walker (Appendix F.2) and Gym Pendulum (Appendix F.3) and observe that MetaPG also discovers a Pareto Front of algorithms that outperform SAC in both environments. Additional information on the stability of the algorithms is presented in Appendix F.4.

Figure 5b compares how the best performer and the best generalizer behave in different instances of the environment in which we change the pole length (all instances form the environment set $\mathcal{E}$ used during evolution). We follow the same procedure described by Dulac-Arnold et al. (2021). The best performer achieves better return in the training configuration than the warm-start's. The best generalizer in turn achieves a lower return but it trades it for higher returns in configurations outside of the training regime, being better at zero-shot generalization. The same behavior holds when using RWRL Walker and Gym Pendulum as training environments (see Appendix F.2 and F.3, respectively).

### 4.3 Transferring evolved algorithms between Brax environments

Figure 6 shows the behaviour of evolved algorithms when meta-tested in perturbed Brax Ant environments (changes in friction coefficient, mass, and torque, see Appendix C). We first evolve algorithms independently in both Ant and Humanoid, then select those algorithms that have the highest stability-adjusted generalizability score $\tilde{f}_{gen}$ during meta-validation. We then re-evaluate using the meta-testing seeds. We compare algorithms evolved in Ant and Humanoid with hyperparameter-tuned SAC.

These results highlight that an algorithm evolved by MetaPG in Brax Ant performs and generalizes better than a SAC baseline. Specifically, we observe a 15% improvement in stability-adjusted performance and 10% improvement in stability-adjusted generalizability. We also obtain a 23% reduction in instability. In addition, we observe that an algorithm initially evolved using Brax Humanoid and meta-validated in Ant transfers reasonably well to Ant during meta-testing, achieving slight loss of performance compared to hyperparameter-tuned SAC (adjusting for stability, 17% less performance and 13% less generalizability compared to SAC). We evolved fewer graphs in the case of Humanoid (50K compared to 200K evolved graphs for Brax Ant), as training a policy in Humanoid is more costly. We expect these results to improve if more algorithms are evolved in the population. We present complete results for the Brax Ant environment in Appendix F.5 and repeat the same analysis for the Brax Humanoid environment in Appendix F.6, in which we observe similar results.

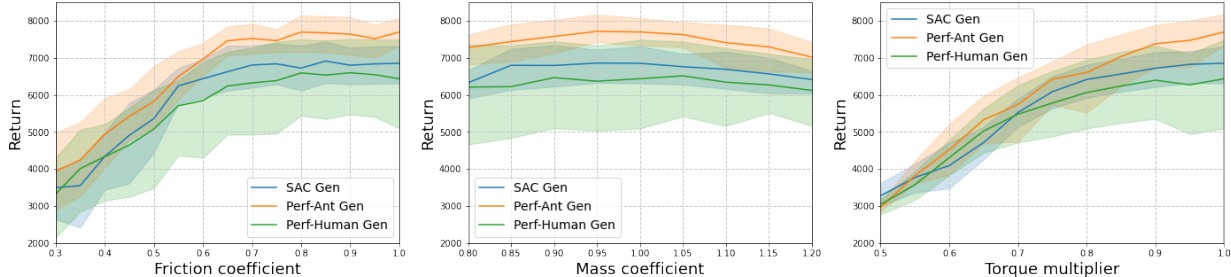

Figure 6: Meta-testing in Brax Ant. We compare, after hyperparameter tuning, loss functions evolved in Brax Ant, Brax Humanoid (to assess cross-domain transfer), and the SAC baseline used as warm-start. We plot the average return and standard deviation across random seeds when evaluating in multiple Brax Ant instances with different friction coefficients, mass coefficients, and torque multipliers (in all cases 1.0 is used as training configuration). The loss function evolved in Ant improves upon SAC's performance and generalizability.

## 4.4 Analyzing the evolved RL algorithms

Next, we analyze evolved algorithms from our experiments on RWRL Cartpole. We pick the best meta-validated performer and generalizer, both evolved from the warm-start SAC (see Appendix D for its graph representation in the search space). The policy loss $L_\pi$ and critic losses $L_{Q_i}$ (one for each critic network $Q_i$, see Appendix A.1 for details) observed from the graph structure for the best performer are the following:

$$L_\pi^{perf} = \mathbb{E}_{(s_t,a_t,s_{t+1})\sim\mathcal{D}} \left[ \log(\min(\pi(\tilde{a}_{t+1}|s_{t+1}), \gamma)) - \min_i Q_i(s_t, \tilde{a}_t) \right] \tag{4}$$

$$L_{Q_i}^{perf} = \mathbb{E}_{(s_t,a_t,r_t,s_{t+1})\sim\mathcal{D}} \left[ \left( r_t + \gamma \left( \min_i Q_{targ_i}(s_{t+1}, \tilde{a}_{t+1}) - \log\pi(\tilde{a}_{t+1}|s_{t+1}) \right) - Q_i(s_t, a_t) \right)^2 \right] \tag{5}$$

where $\tilde{a}_t \sim \pi(\cdot|s_t)$, $\tilde{a}_{t+1} \sim \pi(\cdot|s_{t+1})$, and $\mathcal{D}$ is an experience dataset extracted from the replay buffer. We highlight in blue the changes and additions with respect to SAC, and in red the elements of the SAC loss function that evolution removes. The loss equations for the best generalizer are:

$$L_\pi^{gen} = \mathbb{E}_{(s_t,a_t,s_{t+1})\sim\mathcal{D}} \left[ \log\pi(\tilde{a}_t|s_t) - \min_i Q_i(s_{t+1}, \tilde{a}_t) \right] \tag{6}$$

$$L_{Q_i}^{gen} = \mathbb{E}_{(s_t,a_t,r_t,s_{t+1})\sim\mathcal{D}} \left[ \mathrm{atan} \left( \left( r_t + \gamma \left( \min_i Q_{targ_i}(s_{t+1}, \tilde{a}_t) - \log\pi(\tilde{a}_t|s_t) \right) - Q_i(s_t, a_t) \right)^2 \right) \right] \tag{7}$$

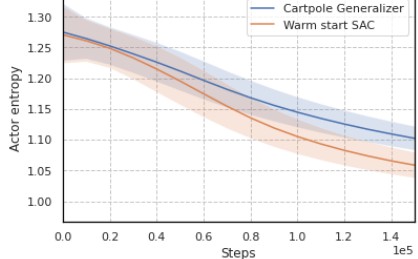

(a) Average entropy of the policy during training for RWRL Cartpole.

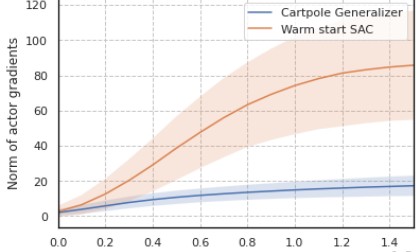

(b) Average gradient norm of the actor loss during training for RWRL Cartpole.

Figure 7: Analysis of the entropy and gradient norm of the actor when evaluating the best generalizer from RWRL Cartpole in comparison to the warm-start. We hypothesize this increase in entropy and decrease in gradient norm with respect to SAC contribute to achieve better generalizability.

While both algorithms resemble the warm-start SAC (see Appendix D), we observe that the best performer does not include the entropy term in the critic loss while the best generalizer does (i.e., they correspond to setting $\alpha$ to 0 and 1 in the original SAC algorithm (Haarnoja et al., 2018), respectively). This aligns with the hypothesis that, since ignoring the entropy pushes the agent to exploit more and explore less, the policy of the best performer overfits better to the training configuration compared to SAC. In contrast, the best generalizer is able to explore more. Figure 7a validates the latter observation showing a higher entropy for the best generalizer's actor compared to the warm-start's.

The use of arctangent in the critic loss of the best generalizer is also noticeable as, supported by Figure 7b, we observe this operation serves as a way of clipping the loss, which makes gradients smaller and thus prevents the policy's parameters from changing too abruptly. In our experiments, we fix the number of training episodes as a compromise between achievable returns and evaluation runtimes. Clipping the loss has then an early-stopping effect compared to the baseline and results in a policy less overfitted, which benefits generalization. In Appendix G.3, we show both extended results that ignore the fix budget requirement and the equations for the best evolved algorithms in the remaining environments. Appendix G.1 and G.2 present the equations for the other RWRL environments and the Brax environments, respectively.

### 4.5 Discussion & Future Work

We have shown that MetaPG can discover novel RL loss functions that achieve better training stability and zero-shot generalization compared to warm-start algorithms such as SAC. Our search space consists only of primitive operators similar to Co-Reyes et al. (2021). While this promotes expressiveness of algorithms in the space, it requires many nodes and edges to represent a loss function which makes our search space vast and good loss functions are extremely sparse in this space. It is challenging for evolutionary algorithms to traverse this space given limited search budget. One area for improvement is to design a more efficient search space so that we have a greater chance of discovering better algorithms under the same computation budget.

Another direction of future work is to improve the transferrability of evolved algorithms to new domains. We have observed that, while evolved algorithms transfer reasonably well (especially best performers, see Appendix G.6), sometimes they do not perform better than SAC in the new environments without hyperparameter tuning first. At the same time, it poses an interesting research question of determining whether MetaPG is better suited to find "super algorithms" for specific environments or a new generation of all-purpose algorithms.

Finally, it would be interesting to ensemble the evolved loss functions on the Pareto front. Such loss function may give additional flexibility for practitioners when designing an RL system by encoding complex design choices into an interpolation across objectives. We hope the released dataset of evolved algorithms help the community gain deeper insights on differences between loss functions.

## 5 Conclusion

We presented MetaPG, a method that evolves actor-critic RL loss functions to optimize multiple RL objectives simultaneously and applied it to discovering algorithms that perform well, achieve zero-shot generalization across different environment configurations, and are stable; a triad of objectives with real-world implications. The experiments in RWRL Cartpole, RWRL Walker, and Gym Pendulum demonstrated that MetaPG discovered algorithms that, when using one environment during evolution and then meta-validating in that same environment, outperform SAC, achieving a 4% and 20% improvement in stability-adjusted performance and stability-adjusted generalizability, respectively, and a reduction of up to 67% in instability. Experiments on Brax Ant and Brax Humanoid proved evolution is successful in more complex environments, achieving a 15% and 10% increase in stability-adjusted performance and stability-adjusted generalizability, respectively. We also observed that, when transferring evolved algorithms to environments different from those used during evolution, the loss of performance and generalizability in the new environment is minimal and is comparable to SAC. Finally, we have analyzed the evolved loss functions and linked specific elements in their structure to fitness results, such as the removal of the entropy term to benefit performance.

**Author Contributions**

Hidden for the double-blind review process.

**Acknowledgments**

Hidden for the double-blind review process.

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
