# OpenReview forum: "Evolving Pareto-Optimal Actor-Critic Algorithms for Generalizability and Stability"
_TMLR — Withdrawn by Authors_

### Review · Reviewer_rZma · 2023-03-06

**Summary Of Contributions:**

The paper presents an automated method for optimizing the loss function of reinforcement learning methods. This is investigated for the soft actor critic online RL method. The optimization is performed by means of an evolutionary algorithm. The optimization goals are performance, transferability (here called generalizability) and stability.

**Audience:**

Yes

**Claims And Evidence:**

Yes

**Requested Changes:**

**Changes that seem necessary to me**
* The benefits of the method and the research should be more clearly stated. Should the proposed method be used to gain knowledge about the design of loss functions? Should the method be used to solve practical tasks? What would these be specifically?
The motivation needs to be more clearly elaborated and executed through concrete application scenarios.

**Changes I think are advisable**
* The paper seems incomplete. The sentence "This dataset may be further analyzed to understand how algorithmic changes affect the tradeoff between different objectives" is unsatisfactory. Wouldn't this paper be the right place to do just that analysis? Especially if the use of the proposed method should be to gain insights into the design of loss functions.
* I would advocate changing the title, as it stands it promises more than is delivered. After all, it is not algorithms in general that are optimized, but loss functions.

**Strengths And Weaknesses:**

**Strengths**
* The approach of tuning not only hyperparemeters of online RL algorithms, but the complete loss function is interesting.

**Weaknesses**
* The benefit of the method and the studies remains somewhat unclear. Should the proposed method be used to gain knowledge about the design of loss functions? Should the method be used to solve practical tasks? Which practical tasks would these be?
* The given motivation "Generalizability and stability are two key objectives for operating reinforcement learning (RL) agents in the real world. Designing RL algorithms that optimize these objectives can be a costly and painstaking process." does not fit well with what the proposed method in the application scenarios that come to my mind. If so many interactions with the environment are possible anyway, as needed for evolutionary optimization of the loss function, why use online RL at all? Why not search directly in policy space (without Q-function)?
* The paper seems somewhat preliminary. The sentence "This dataset may be further analyzed to understand how algorithmic changes affect the tradeoff between different objectives" is unsatisfactory. Wouldn't this paper be the place to do just that analysis? Especially if the benefit of the proposed method should be to gain insights on the design of loss functions.

**Further comments**
* It should be defined earlier and more clearly what is meant by „performance“, „generalizability“, and „stability“.
* I would advocate changing the title, as it stands it promises more than is delivered. After all, it is not algorithms in general that are optimized, but loss functions.
* „a RL“ -> „an RL“
* „rubik“ -> Rubik“
* „mario“ -> „Mario“

---

### Review · Reviewer_QHk5 · 2023-03-13

**Summary Of Contributions:**

The authors call upon a multiobjective optimization method (NSGA-II) to evolve loss functions for actor-critic (AC) algorithms. The loss functions are described with a symbolic language which can give rise to a huge space of loss functions graphs (up to 10^400). They start from SAC, they have an inconsistent graph pruning mechanism, they evaluate approximately 100K graphs using a generalization and a performance measure, plus an implicit stability measure. They find several loss functions that outperform SAC's on a set of 5 environments where they vary some elements to measure generalization. Finally, they analyze two obtained loss functions.

**Audience:**

Yes

**Broader Impact Concerns:**

There is no broader impact statement. I think that the authors should be aware that, if anyone using RL had to go through there optimization method, given its computational cost the impact on the planet might not be sustainable.

**Claims And Evidence:**

No

**Requested Changes:**

To be completely transparent, I already had to review this paper for NeurIPS 2022, where my assessment was negative. So the authors are not lucky to get me again as reviewer. I must mention that the work has improved a little (some details have been clarified, the dataset has been released), but my major concerns remain.

My main point is that the authors need to be more clear about their objective with this paper:

- If their objective is to put forward an AutoML method that anybody else could use for obtaining more efficient actor-critic algorithms (which I assume to be a valid assumption), then they should compare their method with baselines from the related AutoML literature, perform ablations, and better analyze the inner working of their method (see below for more elaborated comments)

- If their objective is more on the application side (they do not focus on the AutoML method itself, but on the better performance one can get with their methodology), then I believe they should address challenging real-world applications that make sense (e.g. a sim-to-real transfer scenario or any application that matters in real life). To me, the fact that using multiobjective optimization can improve performance and generalizability of AC algorithms is no surprise at all, so the purely benchmark-based statement that it can is not interesting enough for a scientific paper (and for justifying the huge computational expense).

- If their objective is more on the analysis of the obtained AC loss functions, then to me the paper falls short of providing a thorough enough analysis and a comparison with other improvements to the SAC algorithm.

Other points:
- the authors design a very large search space (potentially up to 10^400 individuals), but we have no idea how much of these individuals are valid. We know that ~ 100K individuals have been evaluated, but we don't know how much it is with respect to all the available options.
Couldn't it be possible to obtain similar results with a much more restricted search space? Here, ablations and comparisons to baselines would help a lot. This is a major weakness of the paper.

- if the focus is on improving SAC, I believe we learn much more from mathematical analyses than from blind search. See for instance
Vieillard, N., Pietquin, O., & Geist, M. (2020). Munchausen reinforcement learning. Advances in Neural Information Processing Systems, 33, 4235-4246. for changes to RL algorithms similar to the one you obtained, admittedly with more neurons but using much less electric consumption...

- In (4), your algo found it useful to use: min (\pi(a|s), \gamma). \gamma is just a constant here, what sense does it make to clip the proba of action with this constant? Isn't it a lot environment-dependent?

- Similarly, in (5) min_i (Q_{targt_i}(...) ...) does not make sense if you remove min_i -> which i do you take?

More local points:

In abstract: "MetaPG evolves ... without loss of performance". The main text mentions a 15% improvement. Which is right?

Fig. 1 is not much useful, it conveys an obvious message

In "Evolution details/mutation", you describe it as if you were mutating parents: "all individuals are copies of this algorithm’s graph at the beginning. Once the population is initialized, individuals undergo mutations that change the structure of their respective graphs." It would be clearer to say that from the unique original individual you generate N offsprings and you mutate these offsprings. This is a minor detail.


Typos and minor surface issues:

p2, Freeman et al. (2021) -> use citep

In Fig.2 caption: "Each evolved graph is evaluated by training an agent following the algorithm encoded by it" -> the algorithm that it encodes?

- you should not use yellow on a white page

When an equation finishes a sentence, it should come with a dot. This is true at least of Equations 2 and 7.






**Strengths And Weaknesses:**

Strengths:
-

Weaknesses:
-

---

### Review · Reviewer_dvDs · 2023-03-28

**Summary Of Contributions:**

The paper presents MetaPG, an evolutionary algorithm instantiated to search for novel SAC-like actor-critic algorithms. The paper concentrates on the formulation, which takes into account performance, generalization, and stability. The authors present a suite of so-obtained polices which lay on the Pareto front with respect to generalization and performance.

**Audience:**

Yes

**Claims And Evidence:**

Yes

**Requested Changes:**

Please address some (easiest) questions from the list above.

**Strengths And Weaknesses:**

The main strength of the paper is the relevance of the problem undertaken, viz., finding a better offline algorithm. It is also interesting the formulation, which put emphasis on three quantities: performance, generalization, and stability.

The major weakness is hard to understand contributions (or at least it was hard for me).

The method itself, MetaPG, seems to be a fairly straightforward application of an out-of-the-box evolutionary method. This is not a problem per se (and the authors do not claim that it is novel); however, it puts more requirements on the other parts. I would also encourage making a more in-depth comparison to existing work (e.g. [1] seems quite similar). (I acknowledge I am not an expert in this particular field.)

I consider the proposed formulation quite interesting and valuable. However, the results section is somewhat superficial. I would expect:
 - testing more environments (I know that this is costly, I do not insist if there are other strong points)
 - the formulation with zero-shot generalization is quite demanding, I'd love to see some discussion, at least, what would be the case of few-shot generalization
 - the gains in performance and generalization are somewhat modest
 - the gain in stability is very interesting, it is not explained, though, what is its source
 - perhaps the most interesting part, namely, transferring between environments, is tested only in one case. I'd be very interesting to compare what happens in other transfers.
 - I think the paper might also benefit from more analyses, as in Sec 4.4. For example, to answer the question if there is something in common for best-performing algorithms for each env, etc.

The Pareto front, even though advertised in the title, is not studied in the paper. The dataset provides lacks any explanation. In such a form, it cannot be considered as a contribution.

---

### Comment · Reviewer_QHk5 · 2023-03-29
**Author's rebuttal?**

To help me finalize my decision about this paper, I would be glad to see the answer of the authors to my questions.

---

### Note · Authors · 2023-04-01

**Comment:**

After careful considerations, we have decided to withdraw our submission. We agree with the reviewers that our paper could benefit from further work in clarifying the contributions, analyzing the evolved algorithms,  discussing generalizability, etc. We believe making these improvements would take more time than the rebuttal period. We sincerely thank the editor and the reviewers for the time spent reviewing our work and for their improvement suggestions.

**Withdrawal Confirmation:**

I have read and agree with the venue's withdrawal policy on behalf of myself and my co-authors.